# Targeting the Sequences of Circulating Tumor DNA of Cholangiocarcinomas and Its Applications and Limitations in Clinical Practice

**DOI:** 10.3390/ijms24087512

**Published:** 2023-04-19

**Authors:** Kyung-Hee Kim, Hyon-Seung Yi, Hyunjung Lee, Go-Eun Bae, Min-Kyung Yeo

**Affiliations:** 1Department of Pathology, Chungnam National University School of Medicine, Munwha-ro 282, Daejeon 35015, Republic of Korea; 2Laboratory of Endocrinology and Immune System, Chungnam National University School of Medicine, Daejeon 35015, Republic of Korea

**Keywords:** cell-free DNA, circulating tumor DNA, biliary tract cancer, cholangiocarcinoma

## Abstract

Cholangiocarcinoma is a malignant epithelial tumor arising from bile ducts that is frequently fatal. Diagnosis is difficult due to tumor location in the biliary tract. Earlier diagnosis requires less invasive methods of identifying effective biomarkers for cholangiocarcinoma. The present study investigated the genomic profiles of cell-free DNA (cfDNA) and DNA from corresponding primary cholangiocarcinomas using a targeted sequencing panel. Somatic mutations in primary tumor DNA and circulating tumor DNA (ctDNA) were compared and clinical applications of ctDNA validated in patients with cholangiocarcinoma. A comparison of primary tumor DNA and ctDNA identified somatic mutations in patients with early cholangiocarcinomas that showed clinical feasibility for early screening. The predictive value of single-nucleotide variants (SNVs) of preoperative plasma cfDNA positive for somatic mutations of the primary tumor was 42%. The sensitivity and specificity of postoperative plasma SNVs in detecting clinical recurrence were 44% and 45%, respectively. Targetable fibroblast growth factor receptor 2 *(FGFR2)* and Kirsten rat sarcoma virus *(KRAS)* mutations were detected in 5% of ctDNA samples from patients with cholangiocarcinoma. These findings showed that genomic profiling of cfDNA was useful in clinical evaluation, although ctDNA had limited ability to detect mutations in cholangiocarcinoma patients. Serial monitoring of ctDNA is important clinically and in assessing real-time molecular aberrations in cholangiocarcinoma patients.

## 1. Introduction

Cholangiocarcinoma is a malignant epithelial tumor arising from intra- and extrahepatic bile ducts. Although its incidence is relatively low, it is frequently fatal, being responsible for about 13% of all cancer deaths and having a 5-year survival rate of 10% [1,2]. Cholangiocarcinoma is usually asymptomatic in early stages and therefore not diagnosed until advanced stages [1,2]. Radiologic and serologic evaluations of these patients yield results that are often confused with a wide spectrum of inflammatory conditions of the biliary tract, and pathologic diagnosis is difficult due to the tumor location, making it difficult to obtain tissue samples [3]. These limitations may be overcome by the use of less invasive methods of identifying effective biomarkers for cholangiocarcinoma.

Circulating cell-free DNA (cfDNA) and circulating tumor DNA (ctDNA) have shown diagnostic and prognostic value and may substitute for pathologic biopsy and disease-monitoring markers for various cancers, including cholangiocarcinoma [4,5]. cfDNA is DNA released into blood from necrotic and apoptotic cells, including tumor cells [6], and ctDNA is cfDNA originating from tumor cells showing tumor-specific genetic alterations. ctDNA may be an alternative to biopsy as it contains somatic mutations of tumor origin. In addition, ctDNA has been shown to indicate tumor responsiveness to treatment and predict prognosis of cancer patients [7,8,9].

The concordance between somatic mutations in ctDNA and their corresponding primary tumors has been evaluated in several cancers, including cholangiocarcinomas [8,10]. Cholangiocarcinomas share common driver genes including fusion transcripts, protein kinase A pathway, and IDH1/2 mutation, but also show heterogenous mutational signatures depending on their anatomic location and subtypes [11] (Figure 1). *BRAF*, erb-b2 receptor tyrosine kinase 2 *(**ERBB2)*, fibroblast growth factor receptor 2 *(FGFR2)*, and isocitrate dehydrogenase 1 **(***IDH1)* genes have been reported as therapeutically relevant genomic alterations in cholangiocarcinoma patients [12]. Serial analysis of ctDNA has also shown the presence of secondary *FGFR2* mutations, which drive acquired resistance to *FGFR* inhibition [13]. Thus, analysis of ctDNA can be potentially useful in guiding appropriate targeted molecular therapy, in monitoring response to treatment, and in the clinical management of patients with cholangiocarcinoma [13,14].

The present study was designed to assess the clinical utility of targeted next-generation sequencing (NGS) of ctDNA in patients with cholangiocarcinomas and to determine the diagnostic and prognostic role of ctDNA. This study evaluated whether plasma can substitute for tumor tissue in the evaluation of somatic mutations in patients with cholangiocarcinoma by assessing the agreement mutations in primary tumor tissue. In addition, this study assessed whether ctDNA could predict metastasis or recurrence in patients with cholangiocarcinoma by determining changes in ctDNA somatic mutation status during follow-up. The genomic profiles of ctDNA and corresponding primary tumor DNA were determined using a targeted sequencing panel consisting of 118 cancer-related genes, and the clinical application of somatic mutations in ctDNA was validated in patients with cholangiocarcinoma.

## 2. Results

### 2.1. Characteristics of Patients

The baseline demographic and clinical characteristics of the 20 patients are shown in Table 1. These 20 patients consisted of 19 patients with cancers (including 14 with extrahepatic cholangiocarcinoma, 4 with intrahepatic cholangiocarcinoma, and 1 with ampullary carcinoma) and 1 patient with a benign liver abscess. These 20 patients included 17 men and 3 women, ranging in age from 35 to 79 years. Their pathologic stages were classified as I (*n* = 5), II (*n* = 9), III (*n* = 4), and IV (*n* = 1). The stage IV patient (patient 19) did not undergo surgical treatment due to the detection of distant metastases at the time of surgery. Regional lymph node metastases were detected in six patients (patients 4, 8, 9, 12, 13, and 14) at the time of surgery and positive surgical margins (R1) in three patients (patients 8, 12, and 13). Ten patients showed clinically poor outcomes, with three having distant metastases after surgery and seven showing radiologic evidence of recurrence or elevated serum CA19-9 concentrations.

### 2.2. Concentrations of Plasma cfDNA

The mean cfDNA concentration in blood samples from patients with cholangiocarcinoma, as determined using a 4200 TapeStation (Appendix A), was 3.99 ng/μL (range 0.29–37.24 ng/μL), whereas the cfDNA concentration in the patient with a liver abscess was 0.60 ng/μL. Evaluation of patients who underwent surgery showed that the mean postoperative cfDNA concentration (3.43 ng/μL; range 0.29–12.20 ng/μL) was lower than the mean preoperative cfDNA concentration (4.54 ng/μL; range 0.60–37.24 ng/μL), although the difference was not statistically significant (*p* = 0.342). Increased cfDNA levels were not significantly associated with poorer clinicopathologic factors, including increased pathologic stage (*p* = 0.250), lymph node metastasis (*p* = 0.270), or tumor recurrence (*p* = 0.387) (Figure 2). 

### 2.3. Disease-Associated Variants from Pre- and Postoperative ctDNA of Cholangiocarcinoma Patients

Somatic alterations of cfDNA in patients with cholangiocarcinoma were quantitatively analyzed by NGS targeting 118 cancer-related genes. Disease-associated mutations were detected in preoperative peripheral blood from 11 (58%) of 19 patients, with allele frequencies (AF) ranging from 0.1% to 52.42%. The most common disease-associated mutations detected in preoperative ctDNA were in the tumor protein 53 (*TP53)* (5 [26%] of 19 patients), ataxia telangiectasia mutated *(ATM)* (3 [16%] of 19 patients), and cyclin-dependent kinase inhibitor 2A *(CDKN2A)* (2 [11%] of 19 patients) genes, with one patient each having alterations in the Kirsten rat sarcoma virus *(KRAS)*, neurofibromin 1 (*NF1)*, *SMAD4*, Cadherin 1 (*CDH1)*, breast cancer gene 2 (*BRCA2)*, cyclin D2 (*CCND2)*, dihydropyrimidine dehydrogenase *(DPYD)*, UDP glucuronosyltransferase family 1 member a1 (*UGT1A1)*, tyrosine-protein phosphatase non-receptor type 11 (*PTPN11)*, and BRCA1 interacting helicase 1 *(BRIP1)* genes (Figure 3). The most common disease-associated mutations detected in postoperative ctDNA were in the *TP53* (5 [25%] of 20 patients), *ATM* (3 [15%] of 20 patients), checkpoint kinase 2 *(CHEK2)* (2 [10%] of 20 patients), and *BRCA2* (2 [10%] of 20 patients) genes, with one patient each having alterations in the *FGFR2*, *KRAS*, *NF1*, *SMAD4*, *CCND2*, *ERBB2*, anaplastic lymphoma kinase (*ALK)*, tuberous sclerosis 1 *(TSC1)*, epidermal growth factor receptor *(EGFR)*, catenin beta-1 *(CTNNB1)*, and *UGT1A1* genes (Figure 3). CNAs and INDELs, however, were not detected in any of the pre- and postoperative cfDNA blood samples.

Figure 4 shows the overall gene alterations in patients with cholangiocarcinoma subtypes. Analysis of patients with extrahepatic cholangiocarcinoma showed that alterations in *TP53* and *ATM* were the two most common in ctDNA (Figure 4A), whereas *KRAS* was the most common, followed by *APC* and *ATM*, in corresponding tumor tissue (Figure 4B). In patients with intrahepatic cholangiocarcinoma, *TP53* was the most common alteration in ctDNA samples (Figure 4C), whereas *TP53*, *ATM*, *CCND2*, *KRAS*, *BRCA2*, mutS homolog 6 (*MSH6)*, phosphatidylinositol-4,5-bisphosphate 3-kinase catalytic subunit alpha *(PIK3CA)*, *PTPN11*, and *UGT1A1* were detected in corresponding tumor tissue (Figure 4D). Analysis of the patient with ampullary carcinoma showed *BRCA2* and postmeiotic segregation increased 2 (*PMS2)* alterations in the primary tumor but no mutations in cfDNA.

### 2.4. Comparison of Pre- and Postoperative ctDNA and Corresponding Primary Tumors

Somatic mutations in ctDNA and corresponding primary tumor tissues were compared (Figure 5). No disease-associated mutations or CNA were detected in either the ctDNA or tissue from the patient with a benign liver abscess. Of the 19 patients with cholangiocarcinoma, 14 (74%) were positive for somatic tissue mutations, and 11 (58%) were positive for preoperative ctDNA mutations, with AFs ranging from 0.1% to 52.42% (Figure 5A). Both ctDNA and corresponding tumor tissue from five (26%) patients with cholangiocarcinoma were positive for the same disease-associated somatic mutation, whereas both the ctDNA and corresponding tumor tissue from three (19%) patients were negative for somatic mutations. Preoperative ctDNA samples from the remaining nine (47%) patients with cholangiocarcinoma were positive for somatic mutations not detected in the corresponding primary tumors. 

Mutations in postoperative ctDNA were compared with those in corresponding primary tumor tissues from 16 patients. Twelve (75%) tumor samples were positive for somatic mutations, and eleven (69%) were positive for postoperative ctDNA mutations, with AFs ranging from 0.2% to 12.2% (Figure 5B). Of the 16 postoperative ctDNA samples analyzed, 6 (38%) had the same somatic mutations as in corresponding tumor tissue (Figure 5B). Postoperative ctDNA from eight (50%) patients were positive for new somatic mutations not present in the corresponding tumor tissue.

The associations between genetic alterations in ctDNA and the subtypes and stages of cholangiocarcinoma were analyzed (Figure 5). Of the 14 patients with extrahepatic cholangiocarcinoma, 11 (79%) had stages I/II tumors, and 3 (21%) had stages III/IV tumors. Identical somatic mutations in ctDNA and corresponding tumor tissue were present in 3 (27%) of the 11 patients with stages I/II and in 1 (33%) of the 3 with stages III/IV tumors. Of the four patients with intrahepatic cholangiocarcinoma, three (75%) had stages I/II, and one (25%) had a stage III/IV. Identical somatic mutations in ctDNA and corresponding tumor tissue were present in one (33%) of the three patients with stages I/II, but not in the patient with stage III/IV. SNVs in ctDNA and corresponding tumor samples did not correlate with subtypes or stages of cholangiocarcinoma.

### 2.5. Serial Dynamics of ctDNA Mutations 

Patients with stage II and higher cholangiocarcinoma were treated postoperatively with first-line fluoropyrimidine- or gemcitabine-based chemotherapy and were followed-up radiologically and by measuring serum CA19-9 levels to assess their responses to treatment. The diagnostic ability of postoperative ctDNA mutations to predict clinical recurrence was evaluated (Table 2 and Appendix A). Of the nine patients with radiologic evidence of clinical recurrence or metastasis, four (44%) had detectable ctDNA mutation. The sensitivity, specificity, positive predictive value, and negative predictive value of postoperative plasma mutations to detect clinical recurrence were 44%, 45%, 40%, and 50%, respectively (Table 2).

Somatic alterations in cfDNA were quantitatively analyzed to determine postoperative circulating tumor mutational burden over time. Recurrence after complete surgical excision in one patient (Patient 11) was accompanied by new genetic mutations in *CTNNB1* (from 0% at baseline to 0.52% at recurrence), *TSC1* (from 0% to 0.26%), and *ATM* (from 0% to 0.12%) (Figure 6A). Another patient (Patient 12), with ctDNA preoperatively positive for mutations in *TP53* and *CDH1*, was negative for these mutations 1 day after complete surgical excision but showed elevations in VAF of *ERBB2* (from 0.03% to 0.39%), *EGFR* (from 0.26% to 0.83%), *CHEK2* (from 0.16% to 0.83%), and *SMAD4* (from 0.09% to 0.16%) at recurrence, as detected radiologically and by elevated CA19-9 levels (Figure 6B). Another patient (Patient 15) showed VAF increases and decreases for somatic mutations during the course of treatment. Pathologic examination showed positive surgical resection margins followed by recurrence, with a subsequent partial response to chemotherapy (Figure 6C). Serial monitoring of somatic alterations in ctDNA showed dynamic changes from postsurgical examination to recurrence to chemotherapy in VAFs of the *PIK3CA* (from 0% to 7.61% to 0%), *MSH6* (from 0% to 7% to 0%), *PTPN11* (from 2.52% to 1.05% to 0%), *TP53* (from 0.12% to 4.66% to 0.06%), *ATM* (from 0.46% to 0.52% to 0.29%), *CCDN2* (from 0.48% to 0.52% to 0.38%), and *UGT1A1* (from 0.38% to 1.12% to 0.26%) genes. 

## 3. Discussion

Evaluating genetic alterations is essential for the diagnosis and management of solid cancers, especially for selecting targeted therapy. Detection of somatic mutations in ctDNA detection may be a good alternative to conventional tissue biopsy as it is a less invasive method of determining real-time molecular aberrations in patients. The present study found that cholangiocarcinoma-associated ctDNA was detectable in peripheral blood samples from 58% of patients, with rates of mutation detection ranging from 47% to 84% [15,16,17,18]. Mean concentrations of cfDNA were found to be higher, both preoperatively (4.54 ng/μL) and postoperatively (3.43 ng/μL), in patients with cholangiocarcinoma than in a patient with a benign lesion (0.60 ng/μL), significantly differentiating cholangiocarcinoma from benign disease. The increased levels of cfDNA in cholangiocarcinoma patients were not significantly associated with poor clinicopathologic factors, although cfDNA yields were found to positively correlate with larger tumor volume and increased tumor stage [17,19]. Somatic mutations in ctDNA were detected in 25% of patients with early (stage I) cholangiocarcinoma, with rates of detection not differing significantly by pathologic stage. Assessment of mutations in ctDNA may have clinical feasibility in early screening for and detection of bile duct cancers.

The present study also compared disease-associated mutations in pre- and postoperative ctDNAs and corresponding tumor tissues. Preoperative ctDNA may have potential significance as a substitute for invasive biopsy, allowing evaluation of the genetic profiles of tumors without the need for tissue sampling. Compared with the genetic profiles of tumor tissues, the genetic profiles of preoperative plasma cfDNA had a positive predictive value of 42%. New somatic mutations were detected in preoperative ctDNA from 47% of patients with cholangiocarcinoma, suggesting that ctDNA genetic profiling did not fully reflect the somatic mutations present in primary tumors, thereby limiting the clinical application of ctDNA profiles. A comparison of ctDNAs and FFPE samples showed that the rate of concordance of *KRAS* mutations was 42.9% [20]. Moreover, ctDNA was found to have high sensitivity (90–94.7%) and specificity (96.7–99.9%) for predicting tumor DNA SNVs of frequently mutated genes [19,21]. Genetic alterations were found to be dependent on tumor origin (intrahepatic, extrahepatic, or ampulla) [16], target samples (plasma or bile) [18], and assay methods (digital droplet PCR, classic targeted sequencing, or liquid platform) [20,22], with different patterns and frequencies of ctDNA somatic mutations and correlation rates.

The associations of postoperative ctDNA with recurrence and remission can enable postoperative risk stratification and guide decision making for postsurgical patient management. cfDNA mutation abundance was assessed over time in several patients to determine the ability of mutation abundance to detect disease recurrence and metastasis and evaluate response to chemotherapy using ctDNA, even in the absence of radiographic or serologic findings. Serial monitoring and trends showed that ctDNAs had the potential to detect recurrence and metastasis and to evaluate clinical status of patients with cholangiocarcinoma. Of the 16 postoperative ctDNA samples analyzed, 6 (38%) were positive for the same somatic mutations in ctDNA and corresponding tumor tissue, and 8 (50%) were positive for new somatic mutations in postoperative ctDNA in the present study. Postoperative plasma mutations had a sensitivity of 44% and a specificity of 45% in detecting clinical recurrence. Previous studies have shown that ctDNA is useful in the serial monitoring of disease and in determining resistance to targeted therapy [13]. The clinical benefits of chemotherapy may be limited by the emergence of acquired resistance, with alterations in candidate targeted genes detected by ctDNA [23]. Genetic alterations detected in cholangiocarcinomas include fusions and mutations in *FGFR2*, mutations in *IDH1* and *KRAS*, the *BRAF V600E* mutation, and *ERBB2* amplification [15]. Mutations in *FGFR2* have been detected in 5.6% [22] and *FGFR2* and *KRAS* mutations in 5% in the present study of ctDNA samples from cholangiocarcinoma patients, with other targetable aberrations identified in <6% to 44% of cholangiocarcinoma patients [15]. Evaluation of ctDNA may be an option in choosing targeted therapy and in validating novel mutations associated with resistance to targeted therapies.

The present study had several limitations, including the small sample size, retrospective study design, and the inclusion of patients from a single institution. The low concordance of SNV assays of pre- or postoperative ctDNA with corresponding tissue might imply false positive, false negative, or as yet unidentified mutations, limiting the ability to determine the clinical status of cholangiocarcinoma patients. ctDNA analysis included variable shedding of cfDNA depending on the timing of sampling, and variations in the volume and concentration of cfDNA samples. Currently available ctDNA assays may detect only a fraction of somatic mutated genes in primary bile duct tumors. Differences in the genetic profiles of preoperative ctDNA and corresponding tumors, and the differences between pre- and postoperative ctDNA mutational signatures, may have been due to intratumoral heterogeneity or reflect limitations in the ctDNA analysis of plasma samples. Changes in genetic alteration patterns of ctDNA from before to after surgical treatment may represent genetic alterations after primary tumor removal. The presence of mutation-positive ctDNA in patients in remission after surgery may reflect residual ctDNA remaining after surgery, false positive results, ctDNA originating from other precancerous or inflammatory tissues in the bile ducts, or ctDNA originating from other organs. To improve the accuracy of the SNVs detected in the cfDNA, secondary validations including Sanger sequencing, SNaPshot, or SNV-specific ligase methods would be required to confirm mutations and validate function using cfDNA NGS genotyping system. Minimization of contamination issue of cfDNAs, purification and optimal quantification of cfDNA, setting of appropriate control group, and an application of whole genome amplification method could be a solution for this issue and could obtain reliable ctDNA results. Additionally, serial monitoring of the ctDNA would be important in understanding and assessing real-time information in patients with cholangiocarcinoma. The integration of radiologic, serologic, and ctDNA findings can increase understanding of patient status. Further studies of large numbers of patients are required to determine the clinical feasibility of ctDNA using the NGS platform as an application of biomarkers in patients with cholangiocarcinoma.

In conclusion, these findings showed that genomic profiling of cfDNA was useful in determining clinical status and response to treatment in patients with cholangiocarcinoma, but that genomic profiling of ctDNA had clinical limitations. Incorporating ctDNA into clinical trials may allow detection of genetic alterations in response to targeted therapy, further determining the clinical utility of ctDNA as a noninvasive predictive and prognostic biomarker in patients with cholangiocarcinoma. Analysis of ctDNA has enormous potential applications in all stages of cancer management, including in earlier diagnosis. The combination of cfDNA markers with other blood-based markers and advanced radiologic modalities may improve diagnostic accuracy, suggesting that ctDNA has the potential to change current clinical practice by moving from tissue to blood as a source of information.

## 4. Materials and Methods

### 4.1. Patients and Sample Collection

This retrospective study was approved by the Chungnam National University Hospital institutional review board (IRB file no. CNUH 2020-09-015), which waived the requirement for informed consent. All samples were provided by the Biobank of Chungnam National University Hospital, a member of the Korea Biobank Network.

Thirty-six plasma samples obtained from twenty patients between January 2018 and December 2020 were analyzed by the pathology laboratory at the Chungnam National University (Daejeon, Republic of Korea) and by IMBdx, Inc. (Seoul, Republic of Korea). These 20 patients consisted of 19 with cholangiocarcinoma (including 14 patients with extrahepatic cholangiocarcinoma, 4 with intrahepatic cholangiocarcinoma, and 1 with ampullary carcinoma) and 1 with liver abscess. Corresponding primary tumor tissue samples were obtained from the 19 cancer patients, and these tissues were fixed with formalin and embedded in paraffin (FFPE). Preoperative blood samples were collected within 1 week prior to surgery from all enrolled patients; postoperative blood samples were collected at follow-up from 16 of these patients; and serial blood samples were collected from 3 of these 16 patients during 20 months of follow-up. The 1st post-operative samples were collected 3 months after the surgery; the 2nd and the 3rd samples were collected evaluating chemotherapy treatment response. Two licensed pathologists (Yeo and Kim) reviewed patient records and confirmed their diagnoses based on the fifth edition of the World Health Organization (WHO) Classification of Digestive System Tumors [24]. Clinical and pathological data, including patient age, type of surgical treatment, and duration of follow-up, were obtained from the electronic medical record system of Chungnam National University Hospital.

### 4.2. Extraction of Cell-Free DNA

Peripheral blood samples were centrifuged, and the collected plasma was stored in liquid nitrogen. DNA was extracted from plasma samples using QIAamp MinElute Midi Kits (4–10 mL) according to the manufacturer’s instructions, with up to 10 mL plasma processed in a standard 15 mL centrifuge tube. Bead binding buffer, magnetic bead suspension, and proteinase K were added to each sample in the appropriate ratio. Samples were lysed at room temperature, circulating DNA was bound to magnetic beads during end-over-end rotation, and magnetic beads with bound DNA were collected as a pellet on a magnet rack. The supernatant was discarded, bound DNA was eluted from the beads using bead elution buffer, and the beads separated from the pre-eluate containing the DNA by centrifugation. Buffer ACB was added to the pre-eluate to adjust conditions to allow optimal binding of the circulating nucleic acids to the membrane. The pre-eluate was loaded onto a QIAamp UCP MinElute column, with circulating DNA adsorbed onto the silica membrane during centrifugation. The membrane containing adsorbed DNA was dried, and the DNA was eluted with ≥20 μL ultraclean water. The concentration and size distribution of circulating nucleic acids purified were determined by analysis using a TapeStation (Agilent Technologies, Santa Clara, CA, USA).

### 4.3. Targeted Cancer Panel Sequencing

A DNA NGS library was constructed at IMBdx, Inc., using an IMBdx NGS DNA Library Prep Kit and the AlphaLiquid^®^ 100 target capture panel (Appendix A). The targeted gene panel included 118 cancer-related genes, covering all exons of these genes (Appendix A). The Illumina Novaseq 6000 platform (Illumina, San Diego, CA, USA) was utilized for sequencing the captured DNA libraries from plasma and tissue, with a 2 × 150 base pair (bp) paired-end mode, achieving a median depth of 53,742×and 4648×, respectively. The sequencing reads were initially demultiplexed and transformed into unmapped bam files, which contained extracted unique molecular identifier (UMI) sequences. These unmapped bam files were then aligned to the GRCh38 reference genome using the Burrows-Wheeler Aligner (BWA) MEM algorithm version 0.7.17-r1188. PCR duplicates were identified by grouping reads that map to the same position and UMI and were collapsed into single-strand consensus sequence (SSCS) or duplex consensus sequence (DCS) using both fgbio tools (http://fulcrumgenomics.github.io/fgbio/: 10 January 2023) and in-house scripts. The median depth of the consensus sequences from plasma and tissue was 9500× (ranging from 5671× to 17,352×) and 4800×, respectively.

To detect copy number alterations (CNAs), a reference sequencing depth profile was prebuilt for the exonic regions targeted by AlphaLiquid^®^ 100 using 50 normal samples. The CNA for each gene was determined to assess whether the normalized depth profile of each tested sample was significantly higher than that of the reference profile. The single nucleotide variation (SNV) and insertion/deletion polymorphism (INDEL) variants were detected using deep blood (IMBdx) that discriminates noise using precomputed background error rate. To detect copy number alterations (CNAs), a reference sequencing depth profile was prebuilt for the exonic regions targeted by AlphaLiquid^®^ 100 using 50 normal samples. The CNA for each gene was determined to assess whether the normalized depth profile of each tested sample was significantly higher than that of the reference profile.

### 4.4. Identification of ctDNA Variants 

Following primary variant calling of the cfDNA samples, germline mutations and variants considered clonal hematopoiesis of indeterminate potential, based on alterations in peripheral blood mononuclear cell samples, were removed. To filter out noise, contamination, and sequencing errors, cutoffs for ctDNA mutations were applied, consisting of variant allele frequency (VAF) ≥ 0.1% and altered DCS count ≥ 4; however, somatic and germline *BRCA* pathogenic mutations were included. Variant allele depth (reads of the mutants) of the cfDNAs ranged from ×8 to ×838. Unexpected false positives were manually curated by visually comparing longitudinal ctDNA mutation profiles. For CNAs, amplifications with CN ≥4 and gains of CN 2.2–4 were defined along with the statistical criteria (*p* < 0.001). For fusions, paired-end reads with overlapping targeted regions were selected to reduce computing time. Candidate fusion genes were detected by dual fusion caller with Genefuse [10.7150/ijbs.24626] and SViCT [10.1093/nar/gkz067]. High-confident fusion genes consisted of those with two or more reads and a mapping quality of 60, with the predicted transcript expected to be functional. The quantity of ctDNA in a sample was evaluated by determining average VAF, calculated as the mean VAFs of all ctDNA mutations in that sample. ctDNA clearance was defined as the disappearance of all the mutations or CNAs above the cutoff value detected at baseline, and sustained ctDNA clearance was defined as ctDNA clearance in at least two consecutive samples. Identified genetic variants were categorized as “pathogenic”, “likely pathogenic”, “uncertain significance”, “conflicting interpretations of pathogenicity”, “presumed benign”, or “benign”, based on their clinical significance according to ClinVar-indexed variants (National Center for Biotechnology Information, USA) [25]. When assessing the mutation frequencies of individual genes, those categorized as “benign” and “presumed benign,” as well as variants with no ClinVar data of clinical significance, were excluded.

### 4.5. Statistical Analysis

Associations between ctDNA and clinicopathological variables were evaluated using Spearman’s rank correlations, Mann–Whitney U-tests, and Fisher’s exact tests as appropriate. The diagnostic value (sensitivity, specificity, positive predictive value, and negative predictive value) of ctDNA mutations was calculated in reference to mutations in the corresponding tumor and clinical prognosis. All statistical analyses were performed using SPSS version 26.0 for Windows (SPSS, Inc., Chicago, IL, USA) and MedCalc version 19.2.0 for Windows (MedCalc Software, Ltd., Ostend, Belgium).

## Figures and Tables

**Figure 1 ijms-24-07512-f001:**
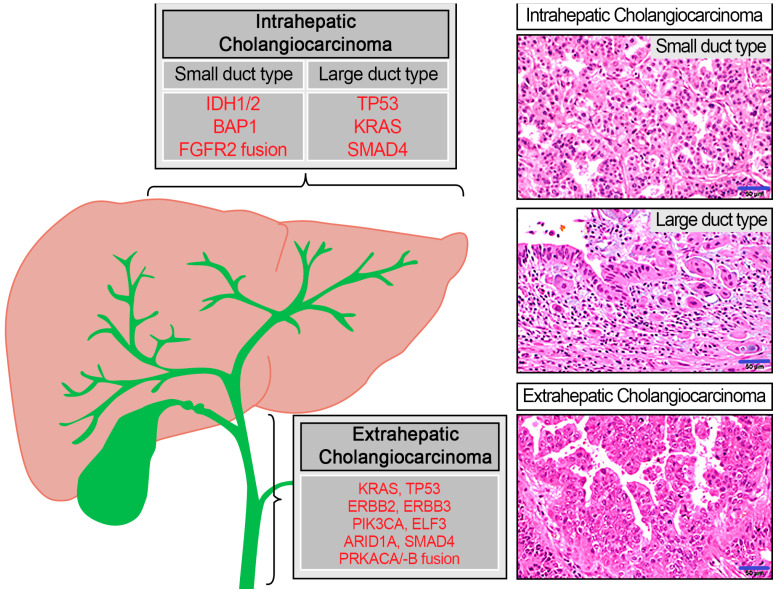
Known driver genes based on subtypes of cholangiocarcinoma and histologic characteristics (Scale bar = 50 μm).

**Figure 2 ijms-24-07512-f002:**
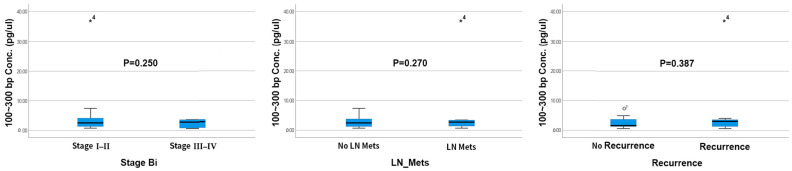
The preoperative plasma cfDNA levels with clinical factors. Mets, metastasis. The case number “*4” showed an extraordinary high concentration level.

**Figure 3 ijms-24-07512-f003:**
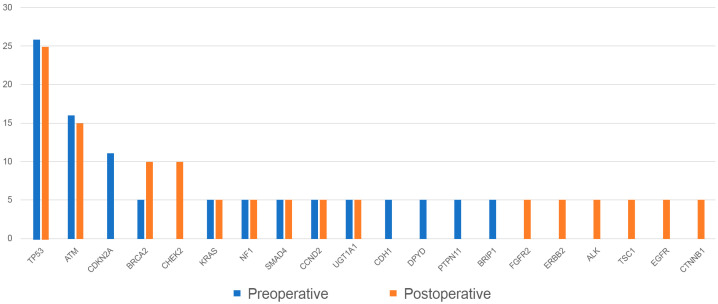
Changes of frequency of alterations of mutated genes in preoperative and postoperative ctDNA samples of patients with cholangiocarcinoma.

**Figure 4 ijms-24-07512-f004:**
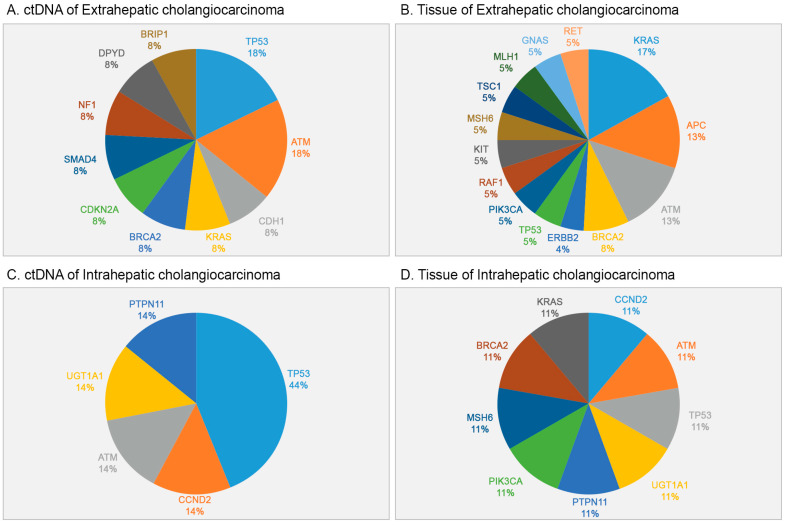
Frequency of alterations of mutated genes in (**A**,**C**) preoperative ctDNA samples and (**B**,**D**) corresponding tumor samples from patients with (**A**,**B**) extrahepatic cholangiocarcinoma and (**C**,**D**) intrahepatic cholangiocarcinoma.

**Figure 5 ijms-24-07512-f005:**
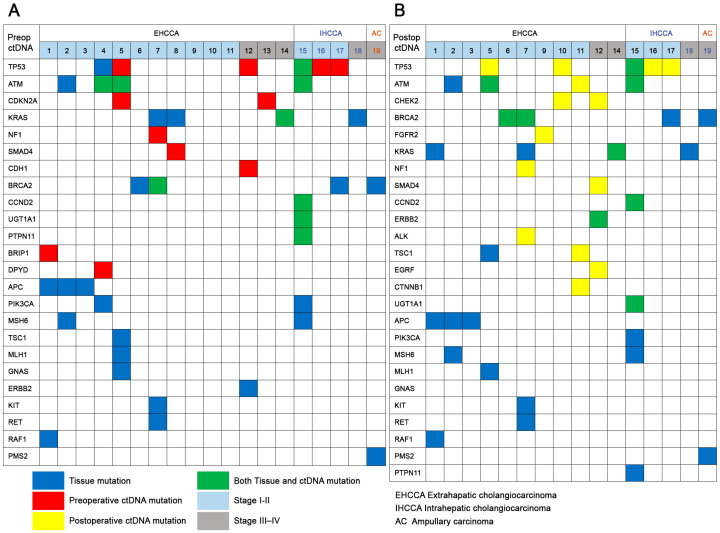
Distribution of somatic mutations in (**A**) preoperative and (**B**) postoperative plasma and corresponding tissue samples.

**Figure 6 ijms-24-07512-f006:**
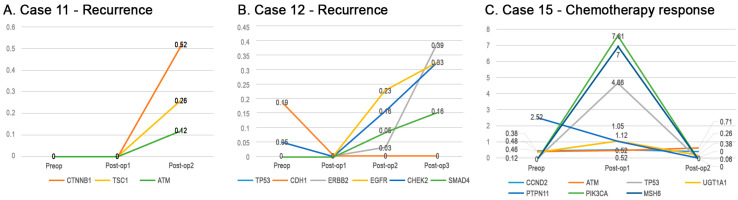
ctDNA detection in patients with mixed clinical responses: disease progression in (**A**,**B**) patients with recurrence and (**C**) a patient who responded to chemotherapy.

**Table 1 ijms-24-07512-t001:** Clinicopathologic characteristics of the patients included in this study (N = 20).

Case	Location	Age	Sex	Diagnosis	TNM	Stage	Diff	Size (cm)	Resection Margin	Outcome
1	Extrahepatic BD	65	M	Adenocarcinoma	pT1 N0	I	wd	2.8	R0	No recur
2	Extrahepatic BD	70	F	Adenocarcinoma	pT2 N0	IIA	md	3	R0	No recur
3	Extrahepatic BD	70	M	Adenocarcinoma	pT1 N0	I	md	2.7	R0	No recur
4	Extrahepatic BD	75	M	Adenocarcinoma	pT2 N1	IIB	md	4.5	R0	Recur
5	Extrahepatic BD	79	F	Adenocarcinoma	pT3 N0	IIB	md	2.5	R0	No recur
6	Extrahepatic BD	69	M	Adenocarcinoma	pT3 N0	IIB	md	2.5	R0	Recur
7	Extrahepatic BD	62	M	Adenocarcinoma	pT1 N0	I	md	1.1	R0	No recur
8	Extrahepatic BD	70	M	Adenocarcinoma	pT2 N1	IIB	md	8	R1	Recur
9	Extrahepatic BD	64	M	Adenocarcinoma	pT2 N1	IIB	md	4.8	R0	No recur
10	Extrahepatic BD	72	M	Adenocarcinoma	pT1 N0	I	md	1.7	R0	No recur
11	Extrahepatic BD	56	M	Adenocarcinoma with neuroendocrine diff	ypT3a N0	IIB	pd	1.1	R0	Mets
12	Extrahepatic BD	69	M	Adenocarcinoma	pT2b N2	IIIA	md	3.2	R1	Recur
13	Extrahepatic BD	75	F	Adenocarcinoma	pT3 N2	IIIA	md	3.8	R1	Recur
14	Extrahepatic BD	74	M	Adenocarcinoma with neuroendocrine diff	pT3a N2	IIIA	pd	4	R0	Mets
15	Intrahepatic BD	75	M	Intrahepatic cholangiocarcinoma	pT1bN0	IB	md	5.9	R0	Recur
16	Intrahepatic BD	62	M	Intrahepatic cholangiocarcinoma	pT2n0	II	pd	5.1	R0	No recur
17	Intrahepatic BD	75	M	Intrahepatic cholangiocarcinoma	pT2 N0	II	md	4.3	R0	No recur
18	Intrahepatic BD	65	M	Intrahepatic cholangiocarcinoma	pT3 N0	IIIA	pd	3	R0	Mets
19	Ampullary area	35	M	Mucinous adenocarcinoma	M1	IV	pd	5.6	NA	Recur
20	Liver	54	M	Abscess	NA	NA	NA	5.8	NA	NA

Diff, differentiation; BD, bile duct; M, male; F, Female; wd, well differentiated; md, moderately differentiated; pd, poorly differentiated; R0, clear resection margin; R1, positive resection margin; NA, not applicable; Recur, recurrence; Mets, metastasis.

**Table 2 ijms-24-07512-t002:** Diagnostic value of SNV of ctDNA by targeted sequencing prediction of clinical recurrence.

Postoperative ctDNA	Clinical Recurrence	No Recurrence	Sensitivity, % (95% CI)	Specificity, % (95% CI)	Positive Predictive Value, % (95% CI)	Negative Predictive Value, % (95% CI)
SNV	4	6				
No SNV	5	5	44.44	45.45	40.0	50.0
Total (*n* = 20)	9	11	(13.70–78.80)	(16.75–76.62)	(21.19–62.31)	(29.48–70.52)

SNV: single-nucleotide polymorphism; CI, confidence interval.

## Data Availability

The raw data were generated at CNUH and derived data supporting the findings of this study are available from the corresponding author on request.

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
