# Peer review of "Targeting the Sequences of Circulating Tumor DNA of Cholangiocarcinomas and Its Applications and Limitations in Clinical Practice"

_ijms, 2023, doi:10.3390/ijms24087512_

Round 1

Reviewer 1 Report

This is a study of the clinical utility of NGS in liquid biopsy samples as an alternative to the use of  tumor tissue to identify somatic mutations, with prediction of outcome. This is a poorly studied tumor type in the context of the liquid biopsy and thus, this gives value to the current study. However, the study has  a limited cohort, only 20 patients. It is true that only surgical cases were included as they study used matched tissue to compare LB and tissue mutations. However, the reliability of the statistical analysis is compromised due to the limited cohort size. Furthermore, the reliability of the cfDNA somatic mutation data is questionable, as many mutations were not validated in the primary tissue. I also have the following specific comments:

The following is stated by the authors: “Preoperative ctDNA samples from the remaining nine (47%) patients with cholangiocarcinoma patients were positive for somatic mutations not detected in the corresponding primary tumors”. This observation is worrying, as they are detecting mutations in cfDNA that cannot be corroborated in the primary tumor. How confident are the authors that the mutations found in cfDNA are real? Can they make some comment abut the read depth and how many reads were of the mutant allele? Along the same lines, the authors state the following: “Postoperative ctDNA from eight (50%) patients were positive for new somatic mutations not present in the corresponding tumor tissue”. Again, it is concerning that they detect mutations in cfDNA that cannot be confirmed in the tumor. At what time after surgery were the samples taken for the post-operative sample? This is not clear from the methods section. If they are follow-up samples taken several months after the surgery, then one may expect to find new mutations that are not present in the original tumor. However, if they were taken within a month or so after surgery, this again would put in doubt the quality of the cfDNA data and the reliability of the mutation detection bioinformatic analysis.

The data shown in section 3.5 is not very reliable due to the low number of patients included in the sensitivity, specificity, positive and predictive value analysis.

The cfDNA concentration data in section 3.2 and supplementary figures would be better shown in a box plot to see the median and spread, even though the analysis was not significant.

It would be more visually informative to combine the separate figures 1a and 1b of the mutation frequency for the pre and post operative samples and use different colors to distinguish the two.

Author Response

Point-by-point response to the comments of the reviewers

Manuscript ID: ijms-2257712

Targeting the Sequences of Circulating Tumor DNA of Cholangiocarcinomas and Its Applications and Limitations in Clinical Practice

 We appreciate the reviewer for taking time to carefully review the manuscript and give detailed and constructive comments, which has helped to improve this paper. Below is our point-by-point response to each comment.

Reviewer #1

 This is a study of the clinical utility of NGS in liquid biopsy samples as an alternative to the use of tumor tissue to identify somatic mutations, with prediction of outcome. This is a poorly studied tumor type in the context of the liquid biopsy and thus, this gives value to the current study. However, the study has a limited cohort, only 20 patients. It is true that only surgical cases were included as they study used matched tissue to compare LB and tissue mutations. However, the reliability of the statistical analysis is compromised due to the limited cohort size. Furthermore, the reliability of the cfDNA somatic mutation data is questionable, as many mutations were not validated in the primary tissue. I also have the following specific comments:

Q1. The following is stated by the authors: “Preoperative ctDNA samples from the remaining nine (47%) patients with cholangiocarcinoma patients were positive for somatic mutations not detected in the corresponding primary tumors”. This observation is worrying, as they are detecting mutations in cfDNA that cannot be corroborated in the primary tumor. How confident are the authors that the mutations found in cfDNA are real? Can they make some comment about the read depth and how many reads were of the mutant allele?

Response: The median depth of the cfDNA was X9,500 (ranged from x5,671 to 17,352) in our study and reads of the mutants of cfDNAs were ranged from x8 to x838. The cutoffs for ctDNA for variant allele frequency (VAF) were ≥0.1% and mutant allele frequency was shown as 5~25% as in the Figure 1. We inserted the sentences in the section 2.3 of the Materials “cfDNA samples were sequenced to a median depth X9,500 (ranged from x5,671 to 17,352) and tissue-derived DNA samples to a median depth x4,8000 revealing a minimal range of genetic alterations” and “Variant allele depth (reads of the mutants) of cfDNAs were ranged from x8 to x838” in the section 2.4 (page 3-4, line 118-121, 146-147).

 Along the same lines, the authors state the following: “Postoperative ctDNA from eight (50%) patients were positive for new somatic mutations not present in the corresponding tumor tissue”. Again, it is concerning that they detect mutations in cfDNA that cannot be confirmed in the tumor. At what time after surgery were the samples taken for the post-operative sample? This is not clear from the methods section. If they are follow-up samples taken several months after the surgery, then one may expect to find new mutations that are not present in the original tumor. However, if they were taken within a month or so after surgery, this again would put in doubt the quality of the cfDNA data and the reliability of the mutation detection bioinformatic analysis.

Response: We collected the 1st post-operative samples 3 months after the surgery and the 2nd and the 3rd samples depending on the chemotherapy treatment responses during the 20 months of follow-ups. We inserted the sentences in the section 2.1 of the Methods “The 1st post-operative samples were collected 3 months after the surgery; the 2nd and the 3rd samples were collected when evaluating chemotherapy treatment response” (page 2, line 88-90).

Q2. The data shown in section 3.5 is not very reliable due to the low number of patients included in the sensitivity, specificity, positive and predictive value analysis.

Response: Our study had several limitations and small sample size was one of the issue. The sensitivity and specificity of our study showed a low diagnostic value as 44% and 45%. We tried to describe the limitation of our study and suggested an integrated analysis with other clinic-radiologic findings for assessing the patient’s status in the discussion section (page 11, page 362-380).

Q3. The cfDNA concentration data in section 3.2 and supplementary figures would be better shown in a box plot to see the median and spread, even though the analysis was not significant.

Response: We changed the supplementary figures to box blot and added as a Figure 1 in the page 5.  

Q4. It would be more visually informative to combine the separate figures 1a and 1b of the mutation frequency for the pre and post-operative samples and use different colors to distinguish the two.

Response: We combined the 1a and 1b and tried to make more visually informative of the changes of the mutation frequency (Figure 2, page 6).

Reviewer 2 Report

The manuscript entitled "Targeting the sequences of circulating tumor DNA of cholangiocarcinomas and its applications and limitations in clinical practice" highlighted that genomic profiling of cfDNA  was useful in clinical evaluation, although ctDNA had limited ability to detect mutations in cholangiocarcinoma patients. Serial monitoring of ctDNA is important clinically and in assessing real-time molecular aberrations in cholangiocarcinoma patients.

- The Authors should provide the expand forms for all acronyms, including gene acronyms, through the text when they first appear.

- Gene acronyms should be written in italics.

Author Response

Reviewer #2

We are grateful to the reviewer for taking time to review the manuscript and give comments. Below is our point-by-point response to each respective comment.

Q1. The manuscript entitled "Targeting the sequences of circulating tumor DNA of cholangiocarcinomas and its applications and limitations in clinical practice" highlighted that genomic profiling of cfDNA was useful in clinical evaluation, although ctDNA had limited ability to detect mutations in cholangiocarcinoma patients. Serial monitoring of ctDNA is important clinically and in assessing real-time molecular aberrations in cholangiocarcinoma patients.

Q1 The Authors should provide the expand forms for all acronyms, including gene acronyms, through the text when they first appear.

Response: We provided the expanded forms of all acronyms including gene names. (page 1, 2, 6, 7)

Q2. Gene acronyms should be written in italics.

Response: We wrote all gene acronyms in italics. 

Reviewer 3 Report

It is recommended to homogenize the columns of table 1 and 2 so that the sentences are not in two lines

homogenize the format of references

Author Response

Reviewer #3

We appreciate to the reviewer for taking time to review the manuscript and give comments. Below is our point-by-point response to each respective comment.

Q1. It is recommended to homogenize the columns of table 1 and 2 so that the sentences are not in two lines

Response: We reduced the font size and homogenized the columns of the Table 1 and Table 2.

Q2. homogenize the format of references

Response: We corrected the ref 17 and homogenized the format of references.

Reviewer 4 Report

Subject: Review report, comments and suggestions for authors

To:

MDPI Office

MDPI IJMS Editorial Office

St. Alban-Anlage 66, 4052 Basel, Switzerland

E-Mail: [email protected]

Sukanya Unson

E-Mail: [email protected]

Journal: International Journal of Molecular Sciences

Recent Advances in Gastrointestinal Cancer

Submitted to section: Molecular Oncology

Title: Targeting the sequences of circulating tumor DNA of cholangiocarcinomas and its applications and limitations in clinical practice

Authors: Kyung-Hee Kim, Go Eun Bae, Hyon-Seung Yi, Hyungjung Lee, Min-Kyung Yeo *

Manuscript ID: ijms-2257712

Title: Targeting the sequences of circulating tumor DNA of cholangiocarcinomas and its applications and limitations in clinical practice

Journal: International Journal of Molecular Sciences

Dear Sirs

Thank you for your request to review ijms-2257712. After careful consideration, I feel that it has merit but does not fully meet your publication criteria as it currently stands. Therefore, I would propose you to submit a revised version of the manuscript that addresses the points raised during my review process.

This is a feasibility study of a novel method for clinical evaluation of cholangiocarcinoma based on genomic profiling of cell-free circulating tumour DNA (cfDNA) isolated from liquid biopsy samples. This approach could facilitate the early, less invasive diagnosis of the tumour that is difficult due to its location in the biliary tract. Authors isolated ctDNA and using next generation sequencing (NGS) successfully identified tumour specific single-nucleotide variants (SNVs) located in 118 cancer-related genes, that could potentially be used as clinical biomarkers of the disease. However, sequence analysis revealed some discrepancies between ctDNA and DNA obtained from primary tumours of the same 20 patients (Figure 2 and 3) that may limit the diagnostic potential of the method, that must be unambiguous even when it comes to a single patient. Even if authors used bigger cohorts (here only 20 samples) to get statistical significance and correlation of selected SNVs biomarkers with the disease in terms of general population, it would not solve the problem of the need to get accurate genotype of the tumour based on the circulating ctDNA, as it can be done with direct biopsy. In addition, the mutation found in ctDNA and not found in tumours may mislead the diagnosis followed by the treatment of each individual patient. Only mutations found in both: tissue DNA and ctDNA (marked in green, Figure 3) can be treated as potential reliable accurate biomarkers reflecting the tumour genotype. Red and yellow squares (figure 3) represent mutations found only in pre- or post-operative ctDNA of samples that cannot reflects the mutations found in tumour tissues (blue suqares). What would be the diagnostic conclusion if autors find CDKN2A mutation only in pre-operative sample of a patient (not find in tissue biopsy)? Taking together false positive results might be a real limitation of the proposed method, however the sources of the false positive results can be identified by various molecular methods and in my opinion the method can be improved to get unequivocal results, that would need further studies that should be done for this paper or at least discussed in the last section of the article.   

Correctly, in the discussion, authors try to find some explanations for these discrepancies suggesting that: The present study had several limitations, including the small sample size, and… limitations of ctDNA analysis included variable shedding of cfDNA depending on the timing of sampling, and variations in the volume and concentration of cfDNA samples. …In addition, assays of preoperative ctDNA may have yielded false positive results or as yet unidentified mutations, limiting the ability to determine the clinical status of cholangiocarcinoma patients. Differences in the genetic profiles of preoperative ctDNA and corresponding tumors, and the differences between pre- and postoperative ctDNA mutational signatures, may have been due to intratumoral heterogeneity or may reflect… Limitations in the ctDNA analysis of plasma samples. In addition, changes in genetic alteration patterns of ctDNA from before to after surgical treatment may represent genetic alterations after primary tumor removal. The presence of mutation-positive ctDNA in patients in remission after surgery may reflect residual ctDNA remaining after surgery, false positive results, ctDNA originating from other precancerous or inflammatory tissues in the bile ducts, or ctDNA originating from other organs. Serial monitoring of the ctDNA may be important in understanding and assessing real-time information in patients with cholangiocarcinoma…. Additional studies of large numbers of patients are required to determine the clinical feasibility of using ctDNA as a marker in patients with cholangiocarcinoma…”Genetic alterations were found to be dependent on tumor origin intrahepatic, extrahepatic, or ampulla) [17], target samples 319 (plasma or bile) [19], and assay methods (digital droplet PCR, classic targeted sequencing, or liquid platform) [21,23], with different patterns and frequencies of ctDNA somatic mutations and correlation rates.”

In my opinion the last sentence is the key to improve the accuracy of the SNVs and limitations of the basic methods might be responsible for false positive and negative results. For instance, NGS is usually done by pyrosequencing, that starts from fragmentation of the nucleic acids to be sequenced followed by synthesis of template strand with the assistance of polymerase enzyme. Pyrosequencing has been used for generating short sequence reads (1-100 nucleotides) that have to be done multiple times to get reliable sequences. It is also known that this method limits the system’s ability to accurately perform longer reads. In brief, pyrosequencing is used rather for pre-screening and have to be done several times because is not accurate as classic Sanger’s sequencing method. The same RT-PCR including digital droplet PCR are not the best choice to clearly identify SNPs or SNVs especially in law quality samples of circulating nucleic acids.

·         To be sure that a mutation really exists, Sanger sequencing or Sanger sequencing based methods (e.g. SNaPshot) and/or SNP/SNV specific ligase method should be used as a confirmatory method,

·         very important is the ctDNA enrichment method that should be controlled by reference DNA (known sequence) mixed with tested fluid and isolated together with the clinical samples – that could test sensitivity as well as possible errors of the system including substituted nucleotides,

·         critically important is also not to have contamination of the “fluids” with whole/intact cells, which DNA may interfere and contaminate free ctDNA, sometimes covering the signal of ctDNAs, This can be tested  using fluid smears and hematoxylin/eosin staining as well as prevented by centrifuging the “fluid” samples before at high speed (> 5000 rpm) 10min and checking the precipitates with the same staining methods.

·         Some of the detected mutations might be germline, encoded by non-tumour DNA, therefore, another control DNA samples from oral swab or nuclear blood cells would help to identify them (normal control genotype of the patient),

·         Authors might need to include “normal” controls (individual samples obtained from healthy volunteers), that might also show the source of the false positive and negative results,

·         As the most critical is probably the purification of ctDNA, comparison of 2-3 different kits (different companies) intended to sort ctDNA would be recommended,

·         As quantity of ctDNA is usually very low, a Whole Genome Amplification method (usually used to amplify single cell genomes) would be useful to get reliable, accruable results that would corresponds with mutations found in tumour tissues.

Taking together, although the proposed method cannot be directly applied for clinical evaluation of cholangiocarcinoma (as it needs major improvement and more controls), I would recommend this manuscript to be published after the revision, including the discussion of the false positive and negative results (what should be done) or after it is done.  The authors need to put more details on the methodology used, especially AlphaLiquid profiling, NGS, and confirmatory studies. Diagram of the method in the main article or supplementary data would also help in understanding of the protocol. Moreover, I would suggest to provide (somewhere at the beginning of the article) histopathologic images to show the disease model (cancer tissues) and provide some known and accepted genetic biomarkers of cholangiocarcinoma in the context of top 1-3 their gene network/protein pathways (eg. String pathways).

Minors:

·         Line 95: "Cells were lysed at room temperature"
                I would suggest the change: "samples were lysed at room temperature" as plasma shouldn’t contain cells but ctDNA.

·         Line 204 BRIP1genes needs space before genes.

·         Line 297 cftDNA

·         Line 309 "pathologic biopsy" not sure if this sentence is correct in medical terms... I would suggest invasive biopsy

Good Luck

Author Response

This is a feasibility study of a novel method for clinical evaluation of cholangiocarcinoma based on genomic profiling of cell-free circulating tumour DNA (cfDNA) isolated from liquid biopsy samples. This approach could facilitate the early, less invasive diagnosis of the tumour that is difficult due to its location in the biliary tract. Authors isolated ctDNA and using next generation sequencing (NGS) successfully identified tumour specific single-nucleotide variants (SNVs) located in 118 cancer-related genes, that could potentially be used as clinical biomarkers of the disease. However, sequence analysis revealed some discrepancies between ctDNA and DNA obtained from primary tumours of the same 20 patients (Figure 2 and 3) that may limit the diagnostic potential of the method, that must be unambiguous even when it comes to a single patient. Even if authors used bigger cohorts (here only 20 samples) to get statistical significance and correlation of selected SNVs biomarkers with the disease in terms of general population, it would not solve the problem of the need to get accurate genotype of the tumour based on the circulating ctDNA, as it can be done with direct biopsy. In addition, the mutation found in ctDNA and not found in tumours may mislead the diagnosis followed by the treatment of each individual patient. Only mutations found in both: tissue DNA and ctDNA (marked in green, Figure 3) can be treated as potential reliable accurate biomarkers reflecting the tumour genotype. Red and yellow squares (figure 3) represent mutations found only in pre- or post-operative ctDNA of samples that cannot reflects the mutations found in tumour tissues (blue suqares). What would be the diagnostic conclusion if autors find CDKN2A mutation only in pre-operative sample of a patient (not find in tissue biopsy)? Taking together false positive results might be a real limitation of the proposed method, however the sources of the false positive results can be identified by various molecular methods and in my opinion the method can be improved to get unequivocal results, that would need further studies that should be done for this paper or at least discussed in the last section of the article.   

Correctly, in the discussion, authors try to find some explanations for these discrepancies suggesting that: The present study had several limitations, including the small sample size, and… limitations of ctDNA analysis included variable shedding of cfDNA depending on the timing of sampling, and variations in the volume and concentration of cfDNA samples. …In addition, assays of preoperative ctDNA may have yielded false positive results or as yet unidentified mutations, limiting the ability to determine the clinical status of cholangiocarcinoma patients. Differences in the genetic profiles of preoperative ctDNA and corresponding tumors, and the differences between pre- and postoperative ctDNA mutational signatures, may have been due to intratumoral heterogeneity or may reflect… Limitations in the ctDNA analysis of plasma samples. In addition, changes in genetic alteration patterns of ctDNA from before to after surgical treatment may represent genetic alterations after primary tumor removal. The presence of mutation-positive ctDNA in patients in remission after surgery may reflect residual ctDNA remaining after surgery, false positive results, ctDNA originating from other precancerous or inflammatory tissues in the bile ducts, or ctDNA originating from other organs. Serial monitoring of the ctDNA may be important in understanding and assessing real-time information in patients with cholangiocarcinoma…. Additional studies of large numbers of patients are required to determine the clinical feasibility of using ctDNA as a marker in patients with cholangiocarcinoma…” “Genetic alterations were found to be dependent on tumor origin intrahepatic, extrahepatic, or ampulla) [17], target samples 319 (plasma or bile) [19], and assay methods (digital droplet PCR, classic targeted sequencing, or liquid platform) [21,23], with different patterns and frequencies of ctDNA somatic mutations and correlation rates.”

In my opinion the last sentence is the key to improve the accuracy of the SNVs and limitations of the basic methods might be responsible for false positive and negative results. For instance, NGS is usually done by pyrosequencing, that starts from fragmentation of the nucleic acids to be sequenced followed by synthesis of template strand with the assistance of polymerase enzyme. Pyrosequencing has been used for generating short sequence reads (1-100 nucleotides) that have to be done multiple times to get reliable sequences. It is also known that this method limits the system’s ability to accurately perform longer reads. In brief, pyrosequencing is used rather for pre-screening and have to be done several times because is not accurate as classic Sanger’s sequencing method. The same RT-PCR including digital droplet PCR are not the best choice to clearly identify SNPs or SNVs especially in law quality samples of circulating nucleic acids

To be sure that a mutation really exists, Sanger sequencing or Sanger sequencing based methods (e.g. SNaPshot) and/or SNP/SNV specific ligase method should be used as a confirmatory method,  very important is the ctDNA enrichment method that should be controlled by reference DNA (known sequence) mixed with tested fluid and isolated together with the clinical samples – that could test sensitivity as well as possible errors of the system including substituted nucleotides, critically important is also not to have contamination of the “fluids” with whole/intact cells, which DNA may interfere and contaminate free ctDNA, sometimes covering the signal of ctDNAs, This can be tested using fluid smears and hematoxylin/eosin staining as well as prevented by centrifuging the “fluid” samples before at high speed (> 5000 rpm) 10min and checking the precipitates with the same staining methods. Some of the detected mutations might be germline, encoded by non-tumour DNA, therefore, another control DNA samples from oral swab or nuclear blood cells would help to identify them (normal control genotype of the patient). Authors might need to include “normal” controls (individual samples obtained from healthy volunteers), that might also show the source of the false positive and negative results. As the most critical is probably the purification of ctDNA, comparison of 2-3 different kits (different companies) intended to sort ctDNA would be recommended. As quantity of ctDNA is usually very low, a Whole Genome Amplification method (usually used to amplify single cell genomes) would be useful to get reliable, accruable results that would corresponds with mutations found in tumour tissues.

Response: We appreciate your careful reviews and comments. We agree with your opinion that we have limitation related to the accuracy of the existence of the mutation. We have tried to emphasize the limitation of this study in the discussion and clarified the method throughout the paper. We agree that confirmative methods should be used and a whole genome amplification method would be useful to get reliable results for a small amount of DNA targets. Even though its limitations, we used a NGS target sequencing platform for liquid biopsy, which could be easily applied to the patients using blood sampling in clinical environments with FDA approval tests for the validation of clinical feasibility. We did not include “normal control” for every test of patient samples, we considered common variants with a variant allele frequency between 40% and 60% identified through tissue biopsy and several time points of liquid biopsy samples and enabled the determination of germline variants.

Taking together, although the proposed method cannot be directly applied for clinical evaluation of cholangiocarcinoma (as it needs major improvement and more controls), I would recommend this manuscript to be published after the revision, including the discussion of the false positive and negative results (what should be done) or after it is done.

Response: Thank you for your valuable comment. We have tried to emphasize the limitation of this study and provided a detailed explanation of possibilities related to false positive and negative ctDNA results in the discussion (discussion, line 370-384). We addressed the limitation of NGS system and suggested a methodologic importance for secondary confirmation and cfDNA quality controls (discussion, line 384-395).

 The authors need to put more details on the methodology used, especially AlphaLiquid profiling, NGS, and confirmatory studies. Diagram of the method in the main article or supplementary data would also help in understanding of the protocol.

Response: Thank you the detailed comments. We have provided more details on the method (Methods, line 123-144) and added a diagram of the NGS process using AlphaLiquid profiling in the Supplementary Figure 1.

Moreover, I would suggest to provide (somewhere at the beginning of the article) histopathologic images to show the disease model (cancer tissues) and provide some known and accepted genetic biomarkers of cholangiocarcinoma in the context of top 1-3 their gene network/protein pathways (eg. String pathways).

Response: Thank to your constructive comments, we have provided histopathologic images of cholangiocarcinomas and added known genetic biomarkers of cholangiocarcinomas in the Figure 1.

Minors:

  • Line 95: "Cells were lysed at room temperature"
    I would suggest the change: "samples were lysed at room temperature" as plasma shouldn’t contain cells but ctDNA.
  • Line 204 BRIP1genes needs space before genes.
  • Line 297 cftDNA
  • Line 309 "pathologic biopsy" not sure if this sentence is correct in medical terms... I would suggest invasive biopsy

Response: Thanks to the careful reviews, typos and other errors have been corrected (line 106, 224, 321, 333).

Round 2

Reviewer 1 Report

I am satisfied that the authors have attempted to address my concerns. However, the small cohort size, low specificity/sensitivity and failure to validate mutations found in the liquid biopsy reduces the impact of the study. Although the data are interesting and publishable. I feel that it is more suitable for a lower impact journal.

Author Response

We appreciate your careful reviews and comments to the revised paper.
